# Memory-assisted prompt editing to improve GPT-3 after deployment

**Aman Madaan** [*], **Niket Tandon** [*][†], **Peter Clark**[†], **Yiming Yang**

Language Technologies Institute, Carnegie Mellon University, Pittsburgh, PA, USA
[†] Allen Institute for Artificial Intelligence, Seattle, WA, USA

{amadaan,yiming}@cs.cmu.edu
{nikett,peterc}@allenai.org

## Abstract

Large LMs such as GPT-3 are powerful, but can commit mistakes that are obvious to humans. For example, GPT-3 would mistakenly interpret "What word is similar to *good*?" to mean a homonym, while the user intended a synonym. Our goal is to effectively correct such errors via user interactions with the system but without retraining, which will be prohibitively costly. We pair GPT-3 with a growing memory of recorded cases where the model misunderstood the user's intents, along with user feedback for clarification. Such a memory allows our system to produce enhanced prompts for any new query based on the user feedback for error correction on similar cases in the past. On four tasks (two lexical tasks, two advanced ethical reasoning tasks), we show how a (simulated) user can interactively teach a deployed GPT-3, substantially increasing its accuracy over the queries with different kinds of misunderstandings by the GPT-3. Our approach is a step towards the low-cost utility enhancement for very large pre-trained LMs.[1]

## 1 Introduction

Language models are now better than ever before at generating realistic content, but still lack commonsense (Bender and Koller, 2020; Marcus, 2021). One failure mode due to a lack of commonsense is in misunderstanding a user's *intent*. The typical remedy of retraining with more data is prohibitive due to the cost and infrastructure requirements. In such cases, even if users repeatedly observe the model making a mistake, there are no avenues to provide feedback to the model to make it more accurate and personalized over time.

Our goal is to allow users to correct such errors directly through interaction, and without retraining by injecting the knowledge required to correct the

---

[*]Equal Contribution
[1]Code and data available at https://github.com/madaan/memprompt

Our memory enhanced GPT-3 implementation.

**User:** What word is similar to *good*?
**GPT-3:** The homonym of good is: wood.
**User:** "Similar to" means "with a similar meaning".
**GPT-3:** Noted *[writes to memory]*
**User:** What word is similar to *surprised*?
**GPT-3:** *[Retrieves and adds to prompt '"Similar to" means "with a similar meaning"'].* The synonym of surprised is: amazed.

Figure 1: This paper enhances GPT-3 performance by looking up questions with a similar intent that received any user feedback. Our approach is simple because only the question in the prompt needs to be updated with relevant feedback, and no retraining is necessary.

model's misunderstanding. Building upon the recent success of injecting commonsense in the input (Lewis et al., 2020; Talmor et al., 2020), we propose a novel approach of injecting knowledge in the input via interactive feedback from an end-user.

Our approach, MEM-PROMPT, pairs GPT-3 with a growing memory of cases where the model misunderstood user's intent and was provided with corrective feedback. This feedback is question dependent, and thus the prompt for each sample is *edited* to adapt to the input. In this sense, our work can be seen as an instance of prompt engineering (Liu et al., 2021b) which involves editing the prompts. Our work adds interactivity to prompt engineering as it involves dynamically updating the prompt for every instance.

Figure 1 presents a sample interaction between a user and GPT-3 that our setup enables. The model was asked for a similar word. However, the model's (incorrect) task understanding **u** was "The homonym of good is". The user can detect such discrepancy between the intended and interpreted task instruction, and can provide feedback **fb** as "*similar to* means *with a similar meaning*",

clarifying that they actually wanted a synonym. Crucially, note that such instructional correction is feasible *even if the user does not know the correct answer to their question*, as they are critiquing the model's understanding of their intent, rather than the answers themselves. Thus, our setup **does not** require the users to be experts at tasks being solved, another advantage of our approach.

Further, it is desirable to have a system that can leverage past feedback on new, unseen examples for prompt-editing. We maintain a memory $\mathcal{M}$ of such feedback as a set of key-value pairs, where the key is a misunderstood question, and the value is the user's feedback to correct that misunderstanding. Given a new question, we check if the model has made a mistake on a similar question earlier, by querying the memory for a similar question. If found, append the corresponding feedback to the question prompt. This mechanism aims to prevent the model from making the same type of mistake twice. This failure-driven reminding mechanism draws inspiration from the theory of recursive reminding in psychology (Jacoby and Wahlheim, 2013), which suggests humans index error corrections in the context in which those errors occurred.

This paper sets out the general architecture and a simple implementation of its components. We then demonstrate the system on four tasks, using simulated user feedback: (1) lexical relations (e.g., antonyms, Figure 1), (2) word scrambling (e.g., anagrams), (3) ethics (with user feedback being the appropriate *class* of ethical consideration, e.g., "it is about cheating", using a small set of categories), and (4) ethics (with user feedback being natural language). We find that in all cases, GPT-3's accuracy significantly increases with time, without retraining, as our approach enables it to use corrective feedback from earlier examples to avoid similar misunderstandings on future examples. Our contributions are thus a general architecture and an implementation showing how user feedback might continuously improve model performance, without retraining, in a few-shot prompt setting.

## 2 Related work

Our method builds upon the recent advances in prompt-tuning and few-shot prompting.

Our use of recalled memories is a form of "prompt engineering", where GPT-3's behavior is modified by adding to the query (prompt) (Le Scao and Rush, 2021). Like others, we use GPT-3 with

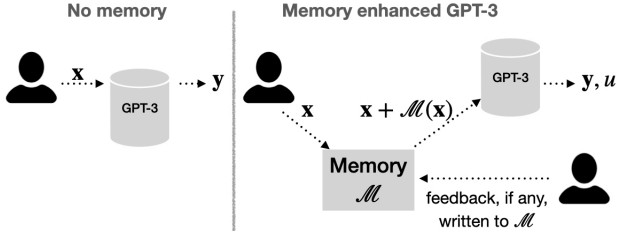

Figure 2: Proposed architecture: (left) GPT-3 does not account for user feedback. (right) MEM-PROMPT maintains a memory $\mathcal{M}$ of corrective feedback, and searches for feedback from prior queries with a similar intent as $x$ using a retrieval function $\mathcal{M}(\mathbf{x})$. $x$ is then concatenated to the retrieved feedback and appended to the prompt for querying GPT-3. Users can also give new feedback on the model's task understanding $u$, then added to $\mathcal{M}$.

*few-shot prompting*, where the prompt consists of a **prefix** $prefix$ containing a few input-output "training" examples of the task, followed by the **input** $x$, e.g., a question, to operate on. However, while prior work has focused on constructing better prefixes, e.g., dynamically selecting good "training" examples based on the question (Liu et al., 2021a), or even representing the prefix latently (Li and Liang, 2021), our work elaborates the input $x$ itself to clarify the intended task, by adding user feedback $fb$ from previous misunderstandings.

Similarly, our work can be seen as a form of retrieval-augmented QA. Extensive prior work has used retrievals from a text corpus to aid QA, e.g., Pan et al. (2019); Guu et al. (2020), or retrievals of prior QA pairs for nearest-neighbor QA (Khandelwal et al., 2020). In contrast, we retrieve from a dynamic memory of user feedback.

The idea of failure-driven reminding and dynamic memory date back several decades, e.g., (Schank, 1983; Riesbeck, 1981). Our work resurrects these ideas in a modern context.

Learning from instruction has become important for large LMs that can perform a task based on direct instruction rather than examples (Wei et al., 2021; Mishra et al., 2021). Our work extends this by adding an adaptive component when those instructions are misinterpreted. While it may not be possible for a user to provide meaningful feedback on the output itself, giving feedback on the understanding of the instruction is more feasible.

Our approach aims to modify the model's behavior through prompting, given a wrong answer. An alternative, recently explored approach is "model editing" - updating the model itself by modifying its parameters to fix incorrect answers (Mitchell

et al., 2021; De Cao et al., 2021; Hase et al., 2021). However, model editing approaches have to date only been demonstrated in a limited context (e.g., correcting a single error), and even then can lead to uncontrollable out-of-scope changes (Mitchell et al., 2021). In contrast, our goal is not just to correct a specific prediction, but to generalize that correction for new problems by collecting feedback to clarify the misunderstanding without damaging the model's basic problem-solving acumen.

Finally, our work is a simple example of debugging and learning via dialog. While system debugging through dialogue has been explored in many contexts (Hixon et al., 2015; Wang et al., 2016; Davis, 1977), our novel contribution is a dialogue about the model's understanding of the user's intent.

# 3 Approach

## 3.1 Memory enhanced GPT-3 architecture

In our setup, given an input $\mathbf{x}$, a model generates an output $\mathbf{y}$ and a sentence $\mathbf{u}$ expressing its understanding of the task, a skill learned through few-shot examples in the prompt (Appendix C). The user can then critique $\mathbf{u}$ by providing natural language feedback $\mathbf{fb}$. This is feasible even if the user does not know the correctness of $\mathbf{y}$ because they are critiquing the *model's understanding of their intent* rather the answers themselves.

Given a new query, MEM-PROMPT uses $\mathbf{fb}$ from similar, prior queries to enrich the (few-shot) prompt $\mathbf{p}$. We use the principle that if two inputs $x_i$ and $x_j$ are similar (i.e., $x_i \sim x_j$), then their feedback $\mathbf{fb}_i$ and $\mathbf{fb}_j$ should be exchangeable ($x_i \sim x_j \Leftrightarrow fb_i \sim fb_j$). The underlying assumption here is that for a fixed model, similar inputs will incur similar errors, and thus can use the same feedback for correction. Fig. 2 gives an overview of MEM-PROMPT, with the following components:

**Memory** $\mathcal{M}$ : $\mathcal{M}$ is a growing table of key ($\mathbf{x}_i$) - value ($\mathbf{fb}_i$) pairs that supports read, write, and lookup operations. The write operation is used whenever a user gives new feedback.

**Lookup** $\mathcal{M}(\mathbf{x})$ : The memory allows lookup operations, denoted as $\mathcal{M}(\mathbf{x})$, that matches the query=$\mathbf{x}$ against all the keys of $\mathcal{M}$.

**Combiner** $\mathcal{C}(\mathbf{x}, \mathcal{M}(\mathbf{x}))$ : A gating function allowing irrelevant, retrieved feedback to be ignored.

**Few-shot prompting** Let us briefly recap few-shot prompting with GPT-3. Consider a general setup where given an input $\mathbf{x}$, a model is expected to generate an output $\mathbf{y}$. In a few-shot prompting mode (Brown et al., 2020), a prompt $\mathbf{p}$ consists of $k$ ($\mathbf{x}, \mathbf{y}$) "in-context" examples, i.e., $\mathbf{p} = \mathbf{x}_1.\mathbf{y}_1\#\mathbf{x}_2.\mathbf{y}_2 \ldots \#\mathbf{x}_k.\mathbf{y}_k$, where $\#$ is a token separating examples and . indicates concatenation. During inference, the user inputs a question $\mathbf{x}_i$, and the model is fed $\mathbf{p} \# \mathbf{x}_i$ (i.e., the question suffixed to the prompt) and is expected to generate the answer $\mathbf{y}_i$ as a continuation.

**MEM-PROMPT setup** As mentioned, given an input $\mathbf{x}$, we prompt the model to generate an output $\mathbf{y}$ and a sentence $\mathbf{u}$ expressing its understanding of the task. Thus, the in-context examples for MEM-PROMPT are of the form $\mathbf{x} \rightarrow \mathbf{u}, \mathbf{y}$. In addition to the input $\mathbf{x}$, MEM-PROMPT retrieves a $\mathbf{fb}$ if a question similar to $\mathbf{x}$ has been asked before. To enable the model to react to such feedback, we also include examples of the form $(\mathbf{x}, \mathbf{fb} \rightarrow \mathbf{u}, \mathbf{y})$ in the prompt, which are aimed to teach the model to react to $\mathbf{fb}$ (Appendix C).

## 3.2 Feedback on model's understanding

In the setup ($\mathbf{x} \rightarrow \mathbf{u}, \mathbf{y}$), there are three modes of failure for a model:

- Task instruction understanding: this is especially concerning in a multi-tasking setup, where the model may consider the question to be about a different task than the one user intended.

- Task nuanced understanding (error on $\mathbf{u}$): when the model understands the task type, but misunderstands the subtle intent in a question.

- Task modeling: if the task is clearly understood, but the answer is not correct, then it requires updating the model parameters. Existing approaches do not scale to very large LMs such as GPT-3, see Section §2 for related work on model editing.

The first two failure modes are due to the inability of the model to understand the input, and are our focus for this work. This paper provides an architecture for a user to critique on model failures. While feedback on the model output is our primary goal, we also experiment with settings where an Oracle is available to provide feedback on the labels (Section §4.3).

We note again that the model reacts to the feed-

| Task (fb type) | (x → y) | u and fb |
|---|---|---|
| Lexical relations (INS) | x: What sounds like good? 
 y: wood | u: Question is asking for a synonym. 
 fb: No, I want a homonym. |
| Word scrambling (INS) | x: Find the right word given this cycled word: elylarg 
 y: largely | u: The question is about anagram. 
 fb: No, its about uncycling a word. |
| Ethical reasoning (CAT) | x: Turning my blender on at 3AM 
 y: It's bad. | u: Question is about authority. 
 fb: No, it is about harm. |
| Ethical reasoning (NL) | x: John has started using again after his mother passed 
 y: It's bad. | u: Question is about spending money. 
 fb: No, it is about drug use. |

Table 1: Feedback types and demonstration of understanding: our system leverages user feedback to prevent failures caused due to a misunderstanding of the task (INS) or semantics of the input (CAT and NL). We achieve this by having the model articulate an understanding u, on which a user can provide feedback using fb.

back because some in-context samples are of the form: $(x, fb → u, y)$ and $(x → u, y)$. We consider a diverse set of tasks $(x → y)$, fb and u, summarized in Table 1.

### 3.3 Tasks

We apply our approach to four tasks: (1) lexical relations (e.g., antonyms, Figure 1), (2) word scrambling (e.g., anagrams), (3) ethics (with user feedback being the appropriate *class* of ethical consideration, and (4) ethics (with user feedback being natural language). For all five tasks, the dataset consists of $(x, fb → u, y)$ tuples, where fb clarifies the task in x. We have a **simulated** conversational setting, in which a user can ask the model x (covering any of these five tasks). If the model gives a wrong answer to query x, then fb is used as the simulated corrective feedback. The sources for these datasets are listed in Appendix §D.

#### 3.3.1 Lexical Relations

The lexical relation task is to predict a word with a given lexical relationship to an input word. We use five relationships: synonym (*syn*), antonym (*ant*), homonym (*hom*, for our experiments, we define homonyms to be the set of words that have different spellings but identical pronunciation, like *ring* and *wring*), definition (*defn*), and sentence usage generation (*sent*).

#### 3.3.2 Word Scrambling

For this task, given a word with its characters transformed, the model is expected to recover the original characters. There are four transformation operations the user can request: reversal of words (*rev*, yppup → puppy), cycle letters in word (*cyc*, atc → cat), random insertions (*rand*, c!r ic/ke!t→ cricket), and anagrams by changing all but the first and last (*anag1*, eelhpnat → elephant) or all but the first and last 2 characters (*anag2*, elapehnt → elephant).

We use the original dataset by Brown et al. (2020).[2]

For both these tasks, each question can be asked in multiple ways (e.g., for synonym generation, the users might ask questions of the form *what is like*, *what has a similar sense*, *what is akin to*, *what is something like*, etc.) Similarly for the lexical relations task, we specify the task description $x$ using different phrasings, e.g., "rearrange the letters" (which the system sometimes misunderstands), and the (simulated) user feedback $fb$ is a clearer task description, e.g., "The anagram is". The system thus accumulates a set of $(x, fb)$ pairs in memory after each failure, helping it avoid future misunderstandings of $x$ through feedback retrieval.

#### 3.3.3 Ethical Reasoning (2 tasks)

For ethical reasoning, we consider a setup where given a situation (e.g., *cheating on your partner*), the model is expected to provide a judgment on whether the situation is ethical or not (e.g., *it's not okay*). In addition to providing a judgment on the ethics of the situation, the model also elucidates its understanding of what the question is about (e.g., *being loyal*). While the user may not know the answer, we posit that they would be able to provide feedback on the broader context. For example, if the model generates *being financially savvy* instead of *being loyal*, a user can still point out this problem and provide feedback.

We use a subset [3] of the dataset provided by DELPHI (Jiang et al., 2021). We simulate two different kinds of user feedback, using two of the annotations attached to each example in the Delphi dataset:

- Categorical feedback (ERT-CAT): In this setting, the model generates its understanding $u$ of the situation by selecting one of 10 different possible

---

[2] word scrambling dataset `https://github.com/openai/gpt-3/tree/master/data`

[3] social norms dataset (social-chemistry-101, Forbes et al. (2020)) `https://github.com/mbforbes/social-chemistry-101`

categories of morality to which the situation might belong: *care, loyalty, authority, fairness, sanctity, degradation, cheating, subversion, betrayal, and harm.* These categories are explicitly provided for each example in the Delphi dataset.

• Natural language feedback (ERT-NL): For this, we use the associated "rule of thumb" (RoT) annotation - a freeform general moral principle - attached to each example in the Delphi dataset. To compile a challenging subset of the data for ERT-NL, we sample by input length, preferring long $\mathbf{x}$, with a short feedback $\mathbf{fb}$. Specifically, we use the top 1% of the inputs by length to create a challenging set of input situations ($\mathbf{x}$). User feedback $\mathbf{fb}$ is a natural language feedback on the understanding $\mathbf{u}$. ERT-NL serves as the most challenging case in our setting. This is in part because our setup relies on the hard problem of retrieving questions that would assume similar feedback. For example, consider two situations: *Filling a false time sheet at work* and *Being at a party, and telling parents I am studying.* These situations look lexically dissimilar but correspond to the same underlying social principle *lying to authority.*

In both the cases, the model is "taught" to generate a category $\mathbf{u}$ (as well as the okay/not-okay answer $\mathbf{y}$ to the ethical question) by being given a few examples in the prompt prefix, thus articulating which moral category (for ERT-CAT) or rule-of-thumb (for ERT-NL) it thinks is applicable. The simulated feedback $\mathbf{fb}$ is the gold category associated with the example in the question, if GPT-3 gets the answer wrong.

We selected these tasks because situations that involve reasoning about similar ethical principles can utilize similar past feedback. For example, *sharing an extra umbrella with your friend if they don't have one*, and *donating surplus food to the homeless* both involve *compassion.*

We note that although the model does not change, adding $\mathbf{fb}$ corrects its erroneous behavior because we provide a few positive "training" examples containing feedback ($\mathbf{x}, \mathbf{fb} \rightarrow \mathbf{u}, \mathbf{y}$) in the prompt (Appendix C).

### 3.4 MEM-PROMPT Implementation

**Implementation of memory** $\mathcal{M}$ $\mathcal{M}$ uses the user input $\mathbf{x}$ as the key and the corresponding feedback $\mathbf{fb}$ as value. Given a question $\mathbf{x}_i$, if the user detects that the model has misunderstood the question, they may provide a $\mathbf{fb}_i$ with probabil-

Figure 3: Sample snapshot of memory for lexical QA.

ity $Pr(\mathbf{f_i})$. The feedback is stored in a memory $\mathcal{M}$, with $\mathbf{x}_i$ as the key and $\mathbf{fb}_i$ as the value. For a subsequent question $\mathbf{x}_j$, the retriever $\mathcal{M}(\mathbf{x})$ (described below) checks if a similar question appears in memory. If yes, then the corresponding feedback is attached with the question and fed to the model for generation.

For example, the model might misunderstand a question asking for synonym, e.g., *what is akin to fast ?* as one that requires antonyms. As mentioned, in our setup, the model generates its understanding of the task $\mathbf{u}$, and not just the answer to the question. The user, by inspecting $\mathbf{u} = $ *The opposite of fast is:* might determine that the model has misunderstood them, and give feedback *i wanted a synonym*, which gets stored in $\mathcal{M}$. If a similar question (e.g., *what is akin to pretty ?*) is asked later by the same or a different user, the corresponding feedback (*i wanted a synonym*) is attached with the question to generate the answer. Figure 3 illustrates a sample memory for this task.

**Implementation of retriever** $\mathcal{M}(\mathbf{x})$ An incorrect retrieved past understanding might cause the model to make a mistake, thus necessitating a good retrieval function. It is hard to have a good retriever because two lexically dissimilar situations can share the same understanding. We found that off-the-shelf methods are insufficient to address these challenges (as shown in Section §4 later). Thus, we propose GUD-IR, a novel two-stage method to look up $\mathbf{x}$ in $\mathcal{M}$. Given a $\mathbf{x}$, GUD-IR first generates a *rough* feedback $\hat{\mathbf{fb}}$ for $\mathbf{x}$ using a generative sequence-to-sequence model. This reduces $\mathcal{M}(\mathbf{x})$ to a search over $\mathbf{fb}_1, \mathbf{fb}_2, \ldots, \mathbf{fb}_{|\mathcal{M}|}$ with $\hat{\mathbf{fb}}$ as the search query. The closest matching entry

is then used as the corresponding **fb**. We defer the details of this method to Appendix A

For the other tasks, we use simpler implementations of $\mathcal{M}(\mathbf{x})$, as the lookup problem was not as challenging as in ERT-NL. In such cases, we rely on existing methods such as Sentence transformers (Reimers and Gurevych, 2019) encoding based retrieval, or heuristics for similarity matching (details in Appendix §E).

**Implementation of combiner** $\mathcal{C}$ $\mathcal{C}$ concatenates **x** with relevant **fb** retrieved by $\mathcal{M}(\mathbf{x})$. To ensure that the **x** is appended with **fb** only if it is relevant, our current implementation of combiner uses a threshold on the similarity score between the **x** and the closest feedback **fb** retrieved by $\mathcal{M}(\mathbf{x})$. We rely on the model (GPT-3) to pay attention to the relevant parts of the input. Exploring more complex gating mechanisms remains an important future work.

## 4 Experiments

**Baselines** We compare our system, MEM-PROMPT (memory-assisted prompt editing) with two different baselines:

- **NO-MEM** This is the standard GPT-3[4] in few-shot prompting mode (hyper-parameters listed in Appendix §B). Input is $\mathbf{p} \# \mathbf{x}_i$ (i.e., question $\mathbf{x}_i$ appended to prompt **p**). It generates answer $\mathbf{y}_i$ and its understanding of the user's intent $\mathbf{u}_i$.
- **GROW-PROMPT:** Similar to NO-MEM, but the **p** is continuously grown with a subset of memory $\mathcal{M}$ that can fit within the prompt (max. 2048 tokens). The most recent subset of $\mathcal{M}$ of memory inserted is inserted in the prompt. The ethical reasoning tasks (ERT) involve long examples, and the initial prompt itself takes close to the max allowed tokens. Thus, the GROW-PROMPT setup is only provided for the lexical relations and word scrambling tasks.

**Metrics** We use two different metrics:

- $Acc(\mathbf{y})$: % of cases where answer matched the ground truth.
- $Acc(\mathbf{u})$: % of cases where the model's understanding of user's intent is correct. $Acc(\mathbf{u})$ is also referred to as instruction accuracy. As discussed in Section §3.2, depending on the task, the model generates its understanding on either the instruction or semantics of the question.

---

[4]We use GPT-3-175B (davinci) for all experiments.

**Clarification probability** In real-world cases, we cannot expect a user to provide feedback for all the examples (the user might not know that the understanding of the model is wrong, for example). To simulate this realistic setting, we experiment with various values of clarification probabilities $c$. Concretely, given an input **x**, and model generated understanding **u**, $c$ is the likelihood of the user providing a feedback **fb** if **u** is wrong..

### 4.1 Main result: MEM-PROMPT improves GPT-3 accuracy

Does pairing GPT-3 with MEM-PROMPT improve performance? Section §4.1.1 empirically validates this question on ethical reasoning tasks and Section §4.1.2 on word reasoning tasks.

#### 4.1.1 Ethical reasoning tasks

Table 2 presents results from running MEM-PROMPT on the DELPHI dataset (1,000 points in the test set). Recall from §3.3 that there are two kinds of feedback on DELPHI questions: CAT and NL feedback. MEM-PROMPT gets over 25% relative improvement for both ERT-NL and ERT-CAT. We found that having an efficient retriever was critical for ERT-NL: sentence transformer based retriever scored 38.5, vs. 45.2 using GUD-IR, a 17% improvement.

| model | ERT-CAT | ERT-NL |
|---|---|---|
| NO-MEM | 48.3 | 34.4 |
| MEM-PROMPT | **60.0** | **45.2** |

Table 2: MEM-PROMPT outperforms NO-MEM for both the categorical and the more challenging ERT-NL setup having longer, ambiguous inputs. Note that ERT-CAT uses sentence transformers for retrieval, whereas ERT-NL uses GUD-IR (Appendix §A).

**MEM-PROMPT effectively incorporates feedback, improving accuracy over time** Figure 5 demonstrates that the instruction accuracy increases over time for different values of clarification probability. The clarification probability $p$ captures the rate at which a user provides feedback on erroneous model understanding. Figure 4 shows the label accuracy also improves over time. The baseline (NO-MEM) quickly saturates after 200 time steps, but MEM-PROMPT continues to improve. The continuous improvement over time is one of the key advantages of our approach. It is also evident from these charts that as instruction accuracy

| Error category | Input | Output (MEM-PROMPT) | Reference |
|---|---|---|---|
| Correct, lexically variant understanding | when I said "i do" to my wife, I am now realizing I only meant "i'll try". does she deserve to know and a chance to back out? | This question is about: telling a spouse your true feelings. | This question is about: loving your partner. |
| Distracted understanding | i am 20 years old, lately I have been forgetting simple spellings while taking notes in class . I often bump into random objects around hurting myself. | This question is about: wanting to kill yourself. The answer is BAD. | This question is about: seeking medical attention if you feel there is something wrong with you. The answer is GOOD. |
| Retrieval failures | checked out my high school ex girlfriends instagram for the first time in two years after graduating. I'm happy I did. she's not as pretty as she used to be and I feel great. | [retrieved feedback] This question is about: looking at someone's social media. The answer is GOOD. | This question is about: judging people based on physical attributes. The answer is BAD. |

Table 3: ERT NL task- error categories

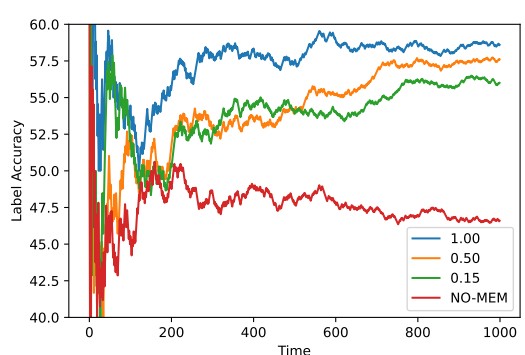

Figure 4: ERT-CAT: Label accuracy increases with time for all values of clarification probabilities.

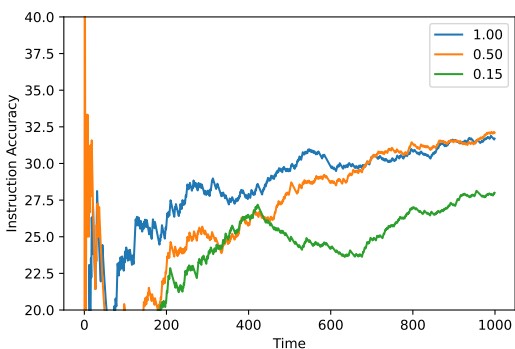

Figure 5: ERT-CAT: Instruction accuracy sharply increases with a larger clarification probability $c$, showing that MEM-PROMPT responds to feedback. With time, lower values of $c$ catch up as memory is gradually filled with error cases and feedback.

improves, the label accuracy also improves (corr. coeff = 0.36).

Further, we also observe that using a higher clarification probability leads to a sharp increase in both instruction and label accuracy early on in the training for both ERT-CAT and ERT-NL. This is because with a large clarification probability, memory fills up with feedback sooner, leading to higher accuracy. While performance with a lower clarification probability lags in the beginning, the accuracy catches up with time as memory is filled with relevant examples.

**Error analysis: Ethical-NL** In both the ERT NL and CAT tasks, one of the primary source of label errors is confusion between labels such as OKAY and GOOD because of the nuanced differences e.g., input = teaching your child a musical instrument. MEM-PROMPT predicted GOOD, while the expected answer was OKAY. Similar trends in this dataset were also observed by Jiang et al. (2021)

We randomly sampled examples from the ERT-NL dev set where the model generates an incorrect understanding (i.e., $Acc(\mathbf{u}) = 0$ based on exact match). Our goal is to understand the typical errors made by the model and use the analysis to calibrate the findings in Table 2. We select ERT-NL for the analysis because it involves free-form natural language which is difficult to study quantitatively.

- **Correct, lexically variant understanding (30%)**: Exact match underestimates the performance of our model (as the task involves generation). $\sim 30\%$ $\mathbf{u}$ is a lexical variation of the reference gold understanding. E.g., *telling a spouse your true feeling* vs. *loving your partner*. Notably, the generated label in these cases is still correct. (Example in Table 3, row 1)
- **Distracted understanding (50%)**: A major source of instruction and label errors is the model getting distracted by an unimportant context. Bad retrieval accounts for 30% errors within this category, e.g., matching a situation in the memory where the expected understanding is only partially applicable to the query. (See Table 3, row 2)

- **Retrieval failures (18%):** These errors are caused by an irrelevant retrieved understanding from the memory, when using a state-of-the-art retrieval method (see Table 3, row 3). With the proposed GUD-IR, we were able to reduce these retrieval failures. See Appendix §A for details..

Canonical examples of these error categories are shown in Table 3. We also find that over time, more relevant past examples are fetched (see Table 7).

### 4.1.2 Word Reasoning Tasks

For these tasks, we compare gold $\mathbf{u}^*$ and generated $\mathbf{u}$ based on some hard-coded linguistic variations (e.g., *the antonym is* matches *the opposite is*). Failure to generate $\mathbf{u}$ is also considered incorrect. While we do not explicitly evaluate the accuracy of the task, we found a near-perfect correlation between the accuracy of $\mathbf{y}$ and $\mathbf{u}$ (i.e., if the GPT-3 understands the task correctly, the output was almost always correct). This shows that if the model is adept at a certain task, improving its understanding of the task might lead to an improved performance.

Figure 6 reports the overall performance on the five lexical tasks overall. The accuracy improves substantially within 300 examples when using memory (in yellow) vs. no memory (in blue). Table 4 breaks down the performance by tasks. We note again that we are operating in a few-shot prompting regime (i.e., there is no training data over which we train). The fact that the model saturates within 300 examples shows that our method can continue to improve. The performance of GROW-PROMPT (red) lies in between, showing that non-selective memory is partially helpful, although not as effective as failure-driven retrieval (our model). However, GROW-PROMPT is $\sim$ 3x more expensive (larger prompts) and cannot scale beyond the 2048 tokens limit. Our model MEM-PROMPT substantially outperforms both the baselines, showing the effectiveness of failure-driven reminding. We also found that the retrieved feedback from memory was effective 97% of the time; only in $\approx$ 3% of cases feedback had no positive effect.

We also note that the performance gains achieved by MEM-PROMPT are less dramatic for word-level tasks. This is explained by the fact that task descriptions for the word scrambling tasks are less ambiguous (Section §3.3), preventing the model from getting confused by users' instructions.

| model | syn | ant | hom | sent | defn | all |
|---|---|---|---|---|---|---|
| NO-MEM | 0.58 | 0.43 | 0.13 | 0.30 | 0.39 | 0.37 |
| GROW-PROMPT | 0.71 | 0.87 | 0.75 | 0.92 | 0.76 | 0.80 |
| MEM-PROMPT | **0.99** | **0.98** | **0.98** | **0.98** | **0.96** | **0.98** |

Table 4: Results lexical QA tasks. Across all tasks, MEM-PROMPT has the best performance.

| model | anag1 | anag2 | cyc | rand | rev | all |
|---|---|---|---|---|---|---|
| NO-MEM | 0.81 | 0.47 | 0.95 | 0.98 | 0.62 | 0.77 |
| GROW-PROMPT | **0.86** | **0.89** | 0.93 | **0.96** | 0.90 | **0.91** |
| MEM-PROMPT | 0.81 | 0.83 | **0.98** | 0.95 | **0.93** | 0.90 |

Table 5: GROW-PROMPT and MEM-PROMPT outperform NO-MEM on all word scramble QA tasks.

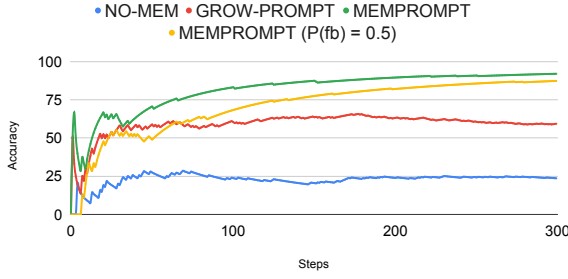

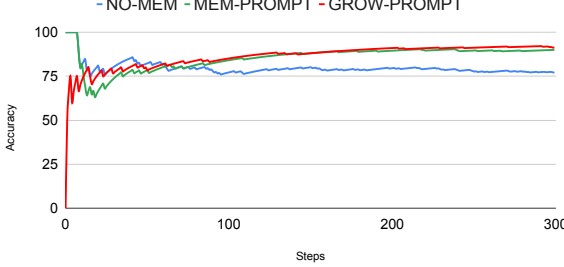

Figure 6: **Main result** Avg. performance on five lexical tasks (top) and word scramble tasks (bottom) with increasing time steps (x-axis). For MEM-PROMPT and GROW-PROMPT, accuracy increases with time as memory is filled up with feedback from past errors.

**Persistent memory use accelerates performance** When the memory is used for every example (green line in Fig 6, top), the performance improves quickly as compared to the yellow line, where $\mathbf{fb}$ from memory is drawn with $Pr(\mathbf{f_i}) = 0.5$.

### 4.2 Using dynamic prefix in prompts

Recent work such as Liu et al. (2021a) investigate using dynamic prompts for better generation. For a given input $\mathbf{x}$, their method( KATE) relies on retrieving examples from the training set that are similar to $\mathbf{x}$ for dynamically creating the prompt $\mathbf{p}$. Note that our method edits $\mathbf{x}$ with a feedback $\mathbf{fb}$, and

is thus complementary to KATE. We experiment with KATE being used to dynamically create the prompt prefix, whereas MEM-PROMPT is used like before to attach a **fb** to the question. We observe a consistent 10% improvement by using KATE across all baselines, verifying our hypothesis that the improvements are complementary.

### 4.3 MEM-PROMPT with label feedback

Our current approach requires the model to verbalize its understanding of the question, on which a user provides feedback. Such a setup might not be possible, for instance, due to the nature of questions. Can MEM-PROMPT be effectively used in such settings as well? To investigate this, we experiment with factual question answering on the WEBQA dataset (Berant et al., 2013), and find clear evidence that MEM-PROMPT is effective even with label feedback (see Appendix §D.3 for details).

### 4.4 Using MEM-PROMPT for language and dialects based personalization

We demonstrate an application of MEM-PROMPT for personalization with a use-case where user language preferences can be folded in the memory. We simulate a user who does not speak fluent English and uses code-mixed language. The queries posed by the user contain words from two Indian languages: Hindi and Punjabi. GPT-3 predictably misunderstands the task. The user clarifies the meanings of their dialect/language phrases. While initial queries fail, subsequent queries that reuse similar words succeed because their clarifications are present in the memory (details in Appendix §E).

## 5 Limitations

**Scaling** We anticipate that practical deployments of MEM-PROMPT like system will use memory as a buffer between cycles of re-training. Concretely, the developers of a model may decide to re-train the model every week. Between cycles of re-training, MEM-PROMPT can serve as a way to avoid repeated mistakes and collect feedback which can be used to fine-tune the model. Depending on the model, a small set of feedback might be enough to capture almost all error classes. In contrast, in some cases, re-training might not be able to fix all the problems of the model. In those cases, memory may help provide the developers with useful examples to improve the underlying model for the next version.

**Quality of feedback** Our setting also assumes that users will not provide adversarial feedback. In real-world settings, this assumption is unlikely to hold. Thus, robust retrieval mechanisms (such as GUD-IR) will be critical for successful real-world deployments.

## 6 Conclusion

We have presented a simple, novel, memory-enhanced GPT-3 that allows users to interact and improve the model without retraining. A key insight is to have the model articulate not just its answer but also its understanding of the user's intent, providing an avenue for feedback. Our implementation of system components are illustrative, not definitive; rather, the goal of this paper is to suggest a general architecture for future researchers, where more sophisticated component implementations can be designed. This architecture is significant as it suggests how deployed systems with fixed models can still be dynamically taught by interacting with end-users, potentially improving their performance and broadening their utility.

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

# A  Generative IR (GUD-IR)

## A.1  Introduction

One of the key strengths of MEM-PROMPT is its ability to leverage feedback provided on earlier inputs $\mathbf{x}$ to improve a current input. This is achieved by retrieving a feedback from memory $\mathcal{M}$ using $\mathbf{x}$ as the key. An underlying assumption of this process is that similar inputs will admit similar feedback, allowing us to use the feedback provided for one situation on another. For two input situations $s_i$ and $s_j$ with respective feedback $\mathbf{fb}_i$ and $\mathbf{fb}_j$, this assumption can be succinctly stated as:

$$s_i \sim s_j \implies \mathbf{fb}_i \sim \mathbf{fb}_j$$

The ethical reasoning dataset with natural language feedback, ERT-NL, fails to meet this assumption because lexically dissimilar situations might have the same feedback, thus posing a unique challenge for our method. As a concrete example, consider an input situation $s_i$: *tom hated skating because he had no sense of balance* – with a feedback $\mathbf{fb}_i$: *practicing more when you want to improve your skills*. Suppose that our system has already seen $s_i$ and has received a feedback $\mathbf{fb}_i$ (i.e., there is an entry in $\mathcal{M}$: $s_i \to \mathbf{fb}_i$). Next, suppose a user enters a new situation $s_j$: *jordyn was trying to improve her soccer skills*. As usual, MEM-PROMPT will try to retrieve feedback for a *similar* situation. However, such retrieval is going to be challenging, because $s_i$ (*tom hated skating because he had no sense of balance*) has little to no overlap with $s_j$ (*jordyn was trying to improve her soccer skills*). Consequently, MEM-PROMPT may fail to retrieve the relevant feedback $\mathbf{fb}_i$ or worse, may retrieve a misleading feedback.

The fact that similarity of two inputs $(\mathbf{x}_i, \mathbf{x}_j)$ does not imply similarity of the feedback $(\mathbf{fb}_i, \mathbf{fb}_j)$ makes vanilla retrieval non-viable for our setting. We deal with this challenging situation with two different solutions of increasing complexity.

## A.2  Initial approach: Fine-tuning with fb similarity

Since the surface level similarity of input situations is not enough to capture similarity of respective feedback, we attempt to learn a function $f_\theta$ that will map similar inputs $\mathbf{x}_i$ and $\mathbf{x}_j$ to similar representations if the corresponding feedback $\mathbf{fb}_i$ and $\mathbf{fb}_j$ are close to each other, and vice-versa. A natural choice is training an embedding function

$f : \mathbf{x} \to \mathrm{R}^d$ supervised by $\cos(\mathbf{fb}_i, \mathbf{fb}_j)$ where $\cos$ is the cosine similarity ($\cos(\mathbf{a}, \mathbf{b}) = \frac{\mathbf{a}^T \mathbf{b}}{|\mathbf{a}||\mathbf{b}|}$). Thus, the objective function is:

$$\mathcal{L}_\theta = (\cos(f_\theta(\mathbf{x}_i), f_\theta(\mathbf{x}_j)) - \cos(\mathbf{fb}_i, \mathbf{fb}_j))^2$$

Intuitively, this objective function will encourage the similarity between the inputs ($\cos(f_\theta(\mathbf{x}_i), f_\theta(\mathbf{x}_j))$) to be high when the corresponding feedback are similar, and vice-versa.

Feedback retrieval proceeds as follows: an input $s_i$ is embedded using $f_\theta$, and $f_\theta(s_i)$ is then used to retrieve a feedback from the memory, with the hope that representations $f_\theta(s_i)$ and $f_\theta(s_j)$ will be similar after the training.

While in principle this objection function should be enough to learn informative representations, we found the training to be unstable. We attribute this to the fact that two extremely dissimilar situations can have identical feedback. Thus, it might be unrealistic to train similarity functions that can capture all possible cases where the same feedback applies to two situations. As a way to circumvent this, we also experiment with a generative version of our method, described next.

## A.3  Proposed approach: Training generative model for retrieving similar feedback

Note from the discussions above that our primary goal is to retrieve a feedback $\mathbf{fb}$ that applies to the given input $s_i$. After observing the unstable training using method discussed in Section §A.2, we experiment with a generative model for the task, described next.

The key intuition for our approach relies on substituting $f_\theta : \mathbf{x} \to \mathrm{R}^d$ with $f_\theta : \mathbf{x} \to \mathbf{fb}$. That is, instead of learning a function that maps a question to a $d$ dimensional vector, we train a generative model that directly maps an input situation a rough feedback. The generated feedback is then used as a key to retrieve a relevant feedback from the training set.

Specifically, we train a sequence-to-sequence model, (e.g., BART or T5), that maps each input $\mathbf{x}$ to a corresponding output $\mathbf{fb}$. The feedback is now retrieved in a two step process:

1. The generative model $f_\theta$ is used to generate a noisy feedback for $s_i$, $\hat{\mathbf{fb}}$.

2. $\hat{\mathbf{fb}}$ is used as a key to *search* over the set of already present feedbacks, to retrieve the nearest one.

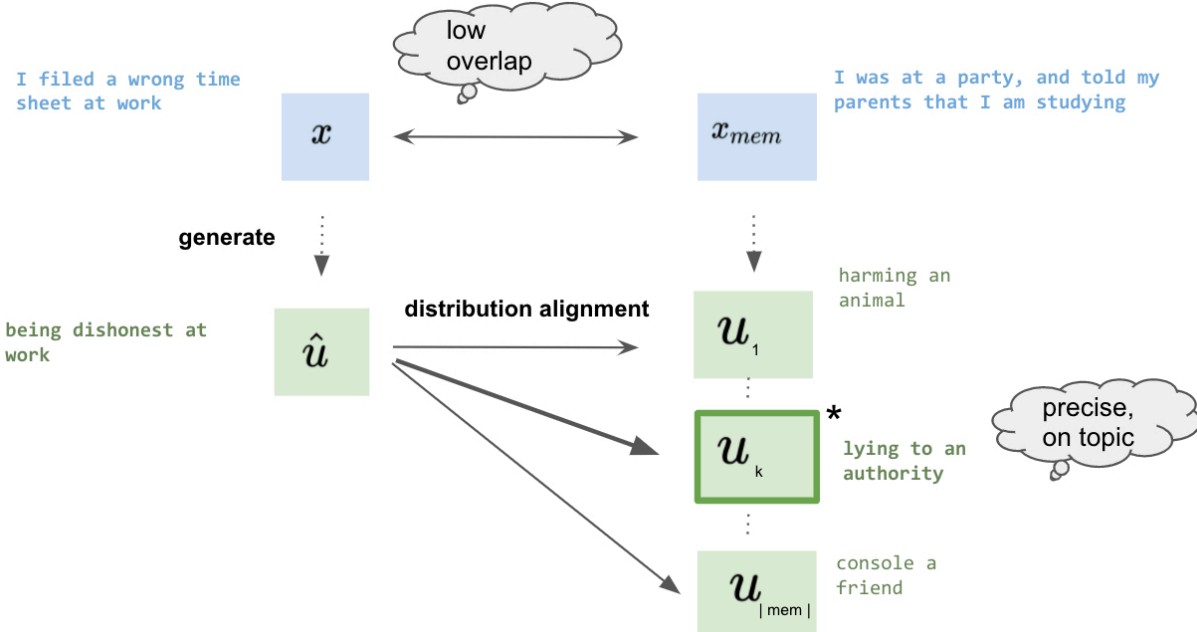

Figure 7: **Overview of GUD-IR.** To retrieve a relevant feedback that applies to $\mathbf{x}$, GUD-IR first generates a feedback $\hat{\mathbf{fb}}$ using a generative model. This is then aligned with a corpus of feedbacks $\mathbf{fb}_1, \mathbf{fb}_2, \ldots, \mathbf{fb}_{|tr|}$ (e.g., sourced from the train split). The best matching feedback $\hat{\mathbf{fb}}^*$ is then used for $\mathbf{x}$. Thus, GUD-IR decomposes the retrieval problem $\mathbf{x} \to \mathbf{fb}$ into two sub-problems: (i) generate a rough feedback ($\mathbf{x} \to \hat{\mathbf{fb}}$) and (ii) search for the closest feedback in a large store $\hat{\mathbf{fb}}^* = \arg\min_{j \in [1, |tr|]} |\hat{\mathbf{fb}} - \mathbf{fb}_j|$.

Intuitively, our generative IR model transforms the lookup problem to a mapping and search problem: instead of directly using clarification to lookup the nearest feedback, we first transform the input to the space of clarifications, then search over the set of already present clarifications. Figure 7 presents an overview of our *generation then reshape* approach (GUD-IR). As we discuss in Section 4.1.1, GUD-IR was key to achieving good performance for the ERT-NL task.

In addition to the task accuracy, we plot the distribution of $\texttt{sim}(\hat{u}, \hat{u}^*)$ (similarity of the true and retrieved feedback) over the test set for different retrieval methods. Figure 8 shows this distribution using GUD-IR and using surface-level similarities. The probability mass shifts towards a higher similarity range for GUD-IR.

The lexical reasoning and WEBQA tasks present a simpler setting for retrieval, as similarity of keys indicates a similarity of values. For such cases, we use Sentence transformers (Reimers and Gurevych, 2019) to encode the query, and cosine similarity with a threshold of 0.9 to find a matching key.

## B  Querying GPT-3-175B using OpenAI API

We use the OpenAI API for querying GPT-3-175B.[5] The python code is listed below. Here, "PROMPT" is set to prompt shown in §C, followed by the input question $\mathbf{x}$ and feedback $\mathbf{fb}$ if applicable.

We used a temperature of 0.0 for factual QA (WEBQA) experiments to select the most likely token at each step, and this setting does not require generating diverse answers, as one would expect for a factual domain. For ERT-CAT and ERT-NL, we found that a higher temperature ($\sim 0.7$) was causing a large divergence in the performance (a difference of $\pm 10\%$ accuracy across runs), making reproducibility challenging – similar observations were made by (Summers-Stay et al., 2021). Thus, we used to a temperature of 0.0 for ERT experiments. A temperature of 0.7 was used for all the other experiments.

```
import os
import openai

openai.api_key = os.getenv("OPENAI_API_KEY")
```

---

[5]https://beta.openai.com/docs/introduction

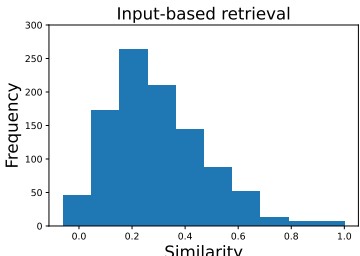 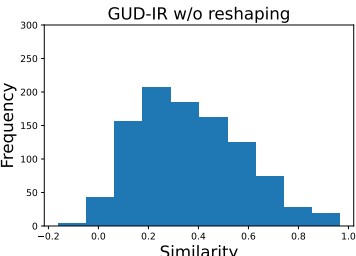 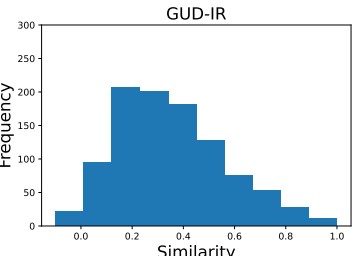

Figure 8: Distribution of similarity scores between expected **fb** $^*$ and $\hat{u}$ for retrieval (left) and GUD-IR (right). The similarity scores are higher using GUD-IR.

```
response = openai.Completion.create(
  engine="davinci",
  prompt="PROMPT",
  temperature=0.7,
  max_tokens=64,
  top_p=1,
  frequency_penalty=0,
  presence_penalty=0
)
```

## C  Prompt

GPT3 is queried using a prompt **p** of example i/o behaviors, followed by the actual question **x** and (optionally) retrieved feedback **fb**. It then generates the understood intent **u** and answer **y** as a continuation. **u** and **y** are expressed a single sentence, e.g., "[The synonym for <word> is] [<word>]" Figure 9 shows this prompt **p**, containing a mixture of $(\mathbf{x} \rightarrow \mathbf{u}, \mathbf{y})$ and $(\mathbf{x}, \mathbf{fb} \rightarrow \mathbf{u}, \mathbf{y})$ "training" tuples.

## D  Datasets for lexical question-answering tasks

As mentioned in Section §4, we focus on five different linguistic QA tasks. The source of data for each of these tasks is listed below:

1. The synonyms (syn) and antonyms (ant) were obtained from Nguyen et al. (2016).[6]

2. The homonyms (hom) were obtained using homz https://github.com/cameronehrlich/homz. We use the closest homonym returned by homz for each word in the English dictionary.

3. The definitions (defn) were sourced from *The Online Plain Text English Dictionary* https://github.com/eddydn/DictionaryDatabase

[6]https://www.ims.uni-stuttgart.de/en/research/resources/experiment-data/lexical-contrast-dataset/

4. Examples for usage in a sentence (sent) are from Commongen (Lin et al., 2020).

### D.1  Templates

We manually created 15 task templates with three variants of phrasing the question for each task. Sample templates are shown in code listing 1. The data (word1, word2) in the code is initialized with the entries in the four sources mentioned above. The complete file is available in the project repository https://github.com/madaan/memprompt/tree/main/src/templates.

### D.2  Sample questions

Tables 8, 9, and 9 list some sample **x**-**y** for settings where the question was asked as a linguistic variation, in Hindi, and in Punjabi, respectively.

### D.3  MEM-PROMPT with label feedback

Our current approach requires the model to verbalize its understanding of the question, on which a user provides feedback. Such a setup might not be possible, for instance, due to the nature of questions. Can MEM-PROMPT be effectively used in such settings as well? To investigate this, we experiment with factual question answering on the WEBQA dataset (Berant et al., 2013), and use the test set provided by Berant et al. (2013) for all experiments (2032 questions). The WEBQA dataset consists of factual questions (*which language is spoken in Canada?*) with multiple answers (*English, French*), and is a popular dataset for benchmarking the performance of GPT-3 on question answering in a few-context prompting setup.

**Inference**  Let $k$ be the number of examples (i.e., question-answer pairs) in the prompt. For a given question $q$, We keep half $(k/2)$ examples fixed in the prompt, whereas the other half $k/2$ examples

```
   What is the homonym for < wring > ?

#
the homonym for wring is ring END
#
how do you use < highway > in a sentence?
#
a sentence with highway is: soldiers stand guard along the [ highway ] END
#
can you define < camisole > ?
#
the definition of camisole is a sleeveless undergarment. END
#
What is the antonym for < prohibition > ?
#
the antonym for prohibition is permit END
#
What is the synonym for < surrogate > ?
#
the synonym for surrogate is substitute END
#
how do i use < fog > ? | clarification: when i ask for how do i use, i want a sentence.
#
a sentence with fog is: a rising sun burns the [ fog ] off a city END
#
What sounds like < sighted > ? | clarification: when I ask for sounds like, I want a homonym.
#
the homonym for sighted is cited END
#
what is like < provident > ? | clarification: when I ask for like, I want a synonym.
#
the synonym for provident is prudent END
#
can you define < rider > ? | clarification: when i ask for define, i want a definition.
#
the definition of rider is a person who is riding something. END
#
What is the opposite of < citation > ? | clarification: when I ask for opposite, I want an antonym.
#
the antonym for citation is award END
```

Figure 9: The prompt used for our tasks. During inference, an input question $\mathbf{x}_i$, and optionally a feedback $\mathbf{fb}_i$ is appended after this prompt, and the model is expected to generate the answer $\mathbf{y}_i$ and its understanding of the question intent $\mathbf{u}_i$ as a continuation. The prompt contains examples of the form $(\mathbf{x} \rightarrow \mathbf{u}, \mathbf{y})$, expressed "$\mathbf{x}$ # $\mathbf{u}$ $\mathbf{y}$ END #", and $(\mathbf{x}, \mathbf{fb} \rightarrow \mathbf{u}, \mathbf{y})$, expressed "$\mathbf{x}$ | clarification: $\mathbf{fb}$ # $\mathbf{u}$ $\mathbf{y}$ END #". ($\mathbf{u}$ and $\mathbf{y}$ are expressed together as a single sentence, e.g., "[The synonym for <word> is] [<word>].")

are retrieved from a memory of feedback $M$. As before, on receiving a question $q$, consults a memory $M$ to see if a similar question has been asked before. However, different from earlier setups, in this case, we retrieve $k/2$ most similar questions from the memory $M$ on **which the system has been wrong earlier**. The corresponding true answers are also retrieved. These $k/2$ retrieved question-answer pairs are combined with the $k/2$ fixed questions to create a prompt, and query GPT-3. Let $a'$

Find the right word after removing random letters from < t!r/e/a/s/u/r.e!s >

#

the word after removing symbols from t!r/e/a/s/u/r.e!s is treasures END

#

Find the original word after ignoring the punctuation and spaces in < e >

#

the word after removing symbols from e is elders END

#

Find the right word given this cycled word: < lprovisiona > ?

#

the uncycled version of lprovisiona is provisional END

#

Make a word while keeping the first and last char < vosiin > ?

#

the anagram 1 for vosiin is vision END

#

Find the original word that is interspersed in < f.i.n!e/p.i/x >

#

the word after removing symbols from f.i.n!e/p.i/x is finepix END

#

Find the right word given this rotated word: < cturalarchite > ?

#

the uncycled version of cturalarchite is architectural END

#

Find the original word after ignoring the punctuation and spaces in < s >

#

the word after removing symbols from s is straightforward END

#

Find the right word given this rotated word: < ibitioninh > ?

#

the uncycled version of ibitioninh is inhibition END

#

Figure out the word which has the same first two and the last two char < watsed > ? | clarification: when I want you to figure out the word which has the same first two and the last two char, I mean anagram 2.

#

the anagram 2 for watsed is wasted END

#

Make a word while keeping the first and last char < isucnase > ? | clarification: when I want you to make a word while keeping the first and last char, I mean anagram 1.

#

the anagram 1 for isucnase is issuance END

#

Unscramble everything except the first two and the last two char < acotrs > ? | clarification: when I want you to unscramble everything except the first two and the last two char, I mean anagram 2.

#

the anagram 2 for acotrs is actors END

Figure 10: The prompt used for our word scrambling tasks derived from Brown et al. (2020)

| % | Question (q) | Example 1 | Example 2 | Example 3 |
|---|---|---|---|---|
| 1 | what highschool did harper lee go to? | what did st augustine do? | who is keyshia cole dad? | when did charles goodyear invented rubber? |
| 4 | what college did albert einstein go to? | what highschool did harper lee go to? | who did tim tebow play college football for? | what timezone is utah in? |
| 40 | where did john mayer go to college? | what school did michael jackson go to high school? | where did derek fisher go to college? | what style of music does john mayer play? |
| 90 | where did scott fitzgerald go to college? | who was f. scott fitzgerald? | where did otto frank go to college? | where did derek fisher go to college? |

Table 6: Relevant examples fetched with time: as time proceeds, the examples fetched from the memory become increasingly relevant to the input question, leading to increasingly accurate predictions.

| Timestep | Question | Neighbor |
|---|---|---|
| 12 | is not wanting a dirty person harassing you morally ok? | is not wanting to hear foul language morally good? |
| 70 | when adam decided to stop living a lie and announces he's gay, are you expected to pretend you're straight when you're gay? | are you expected to pretend you're straight when you're gay? |
| 85 | Should I help someone when they are having trouble? | must you intervene if you see someone taking advantage of another disabled person? |
| 230 | needing a hug to feel good. | is wanting to cuddle with loved ones okay? |

Table 7: Relevant examples retrieved at increasing timesteps: as time proceeds, the examples fetched from the memory become relevant to the input question, leading to accurate predictions.

be the generated answer.

**Growing memory of errors** $M$    In our setup, we assume an expert user (or a teacher) that knows the true answer $a$ for a given query $q$. The expert user compares the GPT-3 generated answer $a'$ with $a$. If the generated answer is correct ($a' = a$), no further action is taken. If not, the entry (($q, a$)) is added to the memory $M$. As time passes, $M$ is populated with an increasing number of challenging examples that the model has been wrong on. Thus, the retrieved $k/2$ examples get more relevant with time, aiding the accuracy. In the experiments, we set $k = 16$ due to budget constraints (note that the setups used in Liu et al. (2021a) and Brown et al. (2020) set $k = 64$, but their results are comparable to our baseline with $k = 16$).

**Results**    Similar to ERT and word reasoning tasks, a memory of errors helps in increasing accuracy with time over 3,000 points in the test split of the WEBQA dataset (Figure 11). This is expected, as $M$ gathers more examples on which GPT-3-175B has been wrong before. Adding these examples in the prompt avoids the model in repeating these mistakes.

To check if examples that belong to a similar domain improve with time, we cluster the questions in the test set of WEBQA, and randomly select three clusters for our analysis. Table 12 shows the top three of the 8 ($k = 16/2$) examples retrieved

from $M$ for the *alma mater* cluster.[7] All of these questions relate to the alma mater of famous personalities. As the inference begins (with an empty $M$), the examples are not relevant to $q$. However, towards the end, almost all the samples are relevant to the given question.

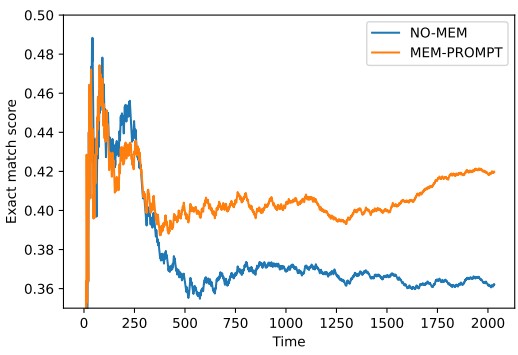

Figure 11: Instruction accuracy vs. time for WEBQA.

## E    Finding similar questions in low-resource settings

We also experimented using queries in Hindi and Punjabi, with (English) feedback clarifying the queries' intent when GPT3 predictably misunderstands the task. Figure 12 confirms significant gains using memory in this OOV setting. This setup

---

[7]Additional examples are included in Appendix §G.

highlights the case when the user does not speak fluent English and uses mixed language code, e.g., transcription in English and mixing words from another language to ask questions.

In low-resource settings (e.g., queries in transcribed Punjabi or Hindi), we perform similarity matching between a given question and a question in the memory by using surface-form similarity. Specifically, we use Levenshtein distance to determine the closest query in the memory. We note that as the memory grows large, we can use mechanisms such as FAISS (Johnson et al., 2019) for trained memory, and suffix-trees for fast retrieval using surface form similarity.

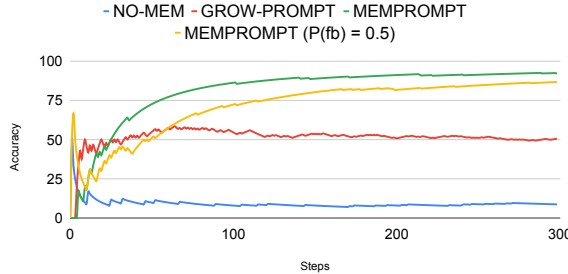

Figure 12: **Finding 2** Large gains on queries asked in English and Punjabi by MEM-PROMPT.

## F Sample results

Table 11 shows randomly sampled **x**-**y** pairs, and the corresponding **y** generated by GPT-3-175B and MEM-PROMPT. The complete set of outputs is located in the project repository `https://github.com/madaan/memprompt/blob/main/results/results.csv`.

# G   Factual question answering

Tables 12 and 13 show additional examples for questions from WEBQA which get additionally relevant examples as time proceeds. The examples include questions that belong to the domains of Alma mater, Soccer, and Language.

```python
templates = [
        {
            "type": "syn",
            "template_id": "syn1",
            "question": lambda word1: f"What is similar to < {word1} > ?",
            "question_clarification": lambda word1: f"What is similar to < {word1} > ? |
    clarification: when I ask for similar to , I want a synonym.",
            "clarification": "clarification: when I ask for similar to , I want a synonym.",
            "answer": lambda word1, word2: f"the synonym for {word1} is {word2}",
        },
        {
            "type": "ant",
            "template_id": "ant0",
            "question": lambda word1: f"What is unlike < {word1} > ?",
            "question_clarification": lambda word1: f"What is unlike < {word1} > ? |
    clarification: when I ask for unlike , I want an antonym.",
            "clarification": "clarification: when I ask for unlike , I want an antonym.",
            "answer": lambda word1, word2: f"the antonym for {word1} is {word2}",
        },
        {
            "type": "defn",
            "template_id": "defn0",
            "question": lambda word: f"< {word} > means what ?",
            "question_clarification": lambda word: f"< {word} > means what ? | clarification:
     when I ask for means what , I want a definition.",
            "clarification": "clarification: when I ask for means what , I want a definition.
    ",
            "answer": lambda word, definition: f"the definition of {word} is {definition}",
        },
        {
            "type": "sent",
            "template_id": "sent1",
            "question": lambda word: f"< {word} > can be used how ?",
            "question_clarification": lambda word: f"< {word} > can be used how ? |
    clarification: when I ask for can be used how , I want a sentence.",
            "clarification": "clarification: when I ask for can be used how , I want a
    sentence.",
            "answer": lambda word, sentence: f"a sentence with {word} is: {sentence}",
        }]
```

Listing 1: Sample templates for the five tasks.

| Question (x) | Answer (y) | type |
|---|---|---|
| What is the opposite of < misconstrue > ? | the antonym for misconstrue is verify | ant |
| What is the opposite of < gross > ? | the antonym for gross is polite | ant |
| expand on < chelicera > ? | the definition of chelicera is One of the anterior pair of mouth organs | defn |
| what is a sentence that can be used to define < mawseed > ? | the definition of mawseed is The seed of the opium poppy. | defn |
| what has a < bitt > like ring to it ? | the homonym for bitt is bit | hom |
| what can one confuse with < holed > ? | the homonym for holed is hold | hom |
| < spread > can be used how ? | a sentence with spread is: a couple of sheep are spread out in a field | sent |
| make something with < pot > ? | a sentence with pot is: bonsai tree in pot at zen garden . | sent |
| What is akin to < musician > ? | the synonym for musician is instrumentalist | syn |
| What is akin to < zigzag > ? | the synonym for zigzag is move | syn |

Table 8: Sample x-y pairs in English. The same type of question can be asked in multiple ways. Our method makes no assumptions as to how a question might be asked.

| Question (x) | Answer (y) | type |
|---|---|---|
| < tabulate > ka ulta kya hai ? | the antonym for tabulate is randomize | ant |
| < foot > ka vilom kya hai ? | the antonym for foot is head | ant |
| < lettish > ka matlab kya hota hai ? | the definition of lettish is The language spoken by the Letts. See Lettic. | defn |
| < housing > ka arth kya hai ? | the definition of housing is An appendage to the hames or collar of a harness. | defn |
| sunne mai < perl > jaisa kya hai ? | the homonym for perl is pearl | hom |
| < council > jaisa kya sunai deta hai ? | the homonym for council is conceal | hom |
| < city > ko ek vakya mai kaise likhen ? | a sentence with city is: the city takes on an even more interesting hue during event | sent |
| < fly > ko ek vakya mai kaise likhen ? | a sentence with fly is: airplane fly into a storm cloud | sent |
| < critique > kai samaan kya hota hai ? | the synonym for critique is evaluate | syn |
| < psychiatric > kai samaan kya hota hai ? | the synonym for psychiatric is mental | syn |

Table 9: Sample **x-y** pairs in Hindi.

| Question (x) | Answer (y) | type |
|---|---|---|
| < edit > de ult ki hunda ae ? | the antonym for edit is expand | ant |
| < spring > ton bhin ki ae ? | the antonym for spring is compress | ant |
| < patricide > di paribhasha dasso ? | the definition of patricide is The crime of one who murders his father. Same as Parricide. | defn |
| < sankhya > da matlab ki hunda ae ? | the definition of sankhya is A Hindoo system of philosophy which refers all things to soul and a rootless germ called prakriti | defn |
| sunnan vich < site > varga ki ae ? | the homonym for site is sites | hom |
| < chance > da samnam ki ae ? | the homonym for chance is chants | hom |
| < enter > nu ek vak vich kidan vartiye ? | a sentence with enter is: as you enter you will be greeted by these rows of trees . | sent |
| < tree > da prayog ki ae ? | a sentence with tree is: wide shot of an eagle and its nest on top of a tree | sent |
| < unlawful > jidan ki hunda ae ? | the synonym for unlawful is illegal | syn |
| < constrict > jidan ki hunda ae ? | the synonym for constrict is close | syn |

Table 10: Sample **x-y** pairs in Punjabi.

| Question (x) | Expected answer (y *) | GPT-3 generated answer | MEM-PROMPT generated answer |
|---|---|---|---|
| give me something i would hear as < whether > ? | the homonym for whether is whither | the example of something you would hear as whether is a question word used to ask about possibilities. | the homonym for whether is weather |
| what has a < mane > like ring to it ? | the homonym for mane is made | the word that has a mane like ring to it is mane | the homonym for mane is main |
| What is like < confine > ? | the synonym for confine is hold | the homonym for confine is confined | the synonym for confine is restrain |
| < leper > means what ? | the definition of leper is a person affected with leprosy. | a sentence with leper is: a leper is a person with a disease that makes his skin scaly and white. | the definition of leper is a person who has leprosy. |
| expand on < chaperon > ? | the definition of chaperon is a matron who accompanies a young lady in public | the expansion of chaperon is one who chaperons. | the definition of chaperon is a woman who goes with a young girl to social functions. |
| what is a sentence that can be used to define < wassail > ? | the definition of wassail is an ancient expression of good wishes on a festive occasion | a sentence with wassail is: < wassail > wassails the christmas season | the definition of wassail is to drink to the health of. |
| expand on < paralbumin > ? | the definition of paralbumin is a proteidlike body found in the fluid from ovarian cysts and elsewhere. it is generally associated with a substance related to | the expansion of paralbumin is an albuminoid globulin occurring in the blood of the horse and other animals, and in the urine of humans. | the definition of paralbumin is a particular kind of albumin. |
| what can one confuse with < kew > ? | the homonym for kew is key | a word that can be confused with kew is due | the homonym for kew is cue |
| What is like < pneumatic > ? | the synonym for pneumatic is windy | the homonym for pneumatic is hydraulic | the synonym for pneumatic is airy |
| give me something i would hear as < seeder > ? | the homonym for seeder is seer | something you would hear as seeder is: someone who plants things in the ground or causes seeds to be planted. | the homonym for seeder is said |

Table 11: A subset of random samples where GPT-3-175B without memory was incorrect. The complete set of outputs is located in the project repository https://github.com/madaan/memprompt/blob/main/results/results.csv

| Domain | % Finished | Question | Neighbor 1 | Neighbor 2 | Neighbor 3 |
|---|---|---|---|---|---|
| Alma mater | 1 | what high-school did harper lee go to? | what did st augustine do? | who is keyshia cole dad? | when did charles goodyear invented rubber? |
| Alma mater | 5 | what college did albert einstein go to? | what high-school did harper lee go to? | who did tim tebow play college football for? | what timezone is utah in? |
| Alma mater | 10 | what university did gordon brown attend? | what all does google now do?' | what team did david beckham play for in 2011?' | who did tim tebow play college football for?' |
| Alma mater | 40 | where did john mayer go to college? | what school did michael jackson go to high school? | where did derek fisher go to college? | what style of music does john mayer play? |
| Alma mater | 75 | where did john steinbeck go to college? | where did john mayer go to college? | what college did john stockton go to? | where did otto frank go to college? |
| Alma mater | 95 | where did scott fitzgerald go to college? | who was f. scott fitzgerald? | where did otto frank go to college? | where did derek fisher go to college? |
| Soccer | 1 | what team did david beckham play for in 2011? | who did tim tebow play college football for? | what super bowl did peyton manning win? | what type of music did john lennon sing? |
| Soccer | 25 | what team did ronaldo play for in 2003? | what part did winona ryder play in star trek? | what to do in richardson dallas? | who did the voice of darth vader in episode 3? |
| Soccer | 33 | who did nasri play for before arsenal? | what year did ray allen join the nba? | who does donnie wahlberg play in the sixth sense? | what does david beckham play? |
| Soccer | 65 | who has pudge rodriguez played for? | who does nolan ryan play for? | who did carlos boozer play for? | who does ronaldinho play for now 2011? |
| Soccer | 99 | what team did david beckham play for before la galaxy? | who does david beckham play for? | what does david beckham play? | what team does david beckham play for in 2012? |

Table 12: Relevant examples retrieved for WEBQA QA task (Section §4.3). The retrieved examples get increasingly relevant as time proceeds.

| Domain | % Finished | Question | Neighbor 1 | Neighbor 2 | Neighbor 3 |
|---|---|---|---|---|---|
| Language | 1 | what does jamaican people speak? | when was ancient egypt created? | where is the denver broncos stadium located? | what is the name of the capital of spain? |
| Language | 20 | what are the two official languages of paraguay? | what do portuguese people speak? | what language does cuba speak? | where is mission san buenaventura located? |
| Language | 37 | what language does colombia? | what language does cuba speak? | what was the first language spoken in spain? | what is serbian language called? |
| Language | 85 | what language does peru speak? | what are the official languages of the eu? | where is the latin language from? | what do portuguese people speak? |
| Language | 90 | what language do they speak in colombia south america? | how many languages do they speak in spain? | where is the latin language from? | what language does cuba speak? |

Table 13: Relevant examples retrieved for WEBQA QA task (Section §4.3). The retrieved examples get increasingly relevant as time proceeds.