# OpenReview forum: "Memory-assisted prompt editing to improve GPT-3 after deployment"
_aclweb.org/ACL/2022/Workshop/CSRR — ACL 2022 Workshop CSRR_

### Official Review · Reviewer_F7Sg · 2022-03-22
**A generally strong paper with some weaknesses to be addressed**

**Rating:** 7
**Confidence:** 4

**Review:**

The presented work proposes a straight-forward method for improving the performance of pre-trained LMs (PLMs) on a variety of tasks through corrective feedback. The feedback is first supplied by a (simulated) user and stored in a memory component. In the main experiments, such feedback contains information that is intended to correct the model's faulty reasoning, e.g. by elaborating on or clarifying the task that the user is expecting the model to perform. The feedback memory can subsequently be used as a source of additional information for new queries to the model, whereby corrective feedback is retrieved from the memory that was previously provided for queries that are most similar to the current one. The query and the retrieved feedback are subsequently combined with the task-specific prompt and given to the PLM as input, which has empirically been found to outperform baselines that either do no utilize the corrective memory at all, or employ a non-selective memory module. The experimental section includes lexical as well as ethical reasoning tasks, although the authors also describe the application of the proposed method to code-switched question answering and question answering with label feedback (as opposed to natural language corrections).

Overall. the proposed method is compelling, easy to implement, and effective according to the provided experimental evaluation. The corrective feedback memory is a neat idea for leveraging and re-using user feedback in an efficient manner and could potentially be applied to many diverse tasks.

There are some minor issues with the paper that should be corrected:
- Line 180: The notation is confusing  - does x_i/j represent the input or the error?
- Line 308: It would be appropriate to provide citation for Social Chemistry 101, as well.
- Figure 3: Some of the example questions / templates are oddly phrased, e.g. "on the lines of" which should probably be either "in the vein of" or "along the lines of"? Could the authors clarify?
- Line 379: I assume this should be "cosine similarity" rather than "cosine distance", as the latter with a threshold of 0.9 would be extremely permissive.
- Line 385: Concatenation is not a gating function (while gating may be explored in the future, it is not part of the presented approach); relatedly, the breakdown of the approach in section 3.1. makes it sound more complex than it actually is and is detrimental to the paper's clarity (e.g. both the "prompter" and the "combiner" are simple concatenation steps and don't really need extra terminology attached to them).
- Line 428: improves -> improve
- Table 2: Should be positioned under 4.1.1 header and does not have to include the GROW-PROMPT row.
- Figure 4: Should get a better caption, e.g. one that explains that the legend denotes the likelihood of feedback being drawn from memory, as it does not to be explicitly stated anywhere else.
- Figure 5: The authors should address why feedback retrieval likelihood of 0.5 performs similar to or better than 1.0.
- Figure 6, caption: Should be "for GROW-PROMPT and MEM-PROMPT".

In closing, I think this paper is really neat, fits well within the body of commonsense reasoning research, and would be a worthy addition to the workshop.

---

### Official Review · Reviewer_bLgH · 2022-03-23
**Solid paper with small writing issues.**

**Rating:** 9
**Confidence:** 4

**Review:**

### Pros

* The paper presents a simple method that works.
* The proposed method does not require model re-training which would be expensive.
* The proposed method supports a natural user-machine interaction.

### Cons

* One downside of the paper is that it only studies the GPT-3 model. It would have been interesting to see if the results apply to the open-source equivalents: GPTNeo and GPTJ.

### Minor things

* Lines 127, 163: The citations should not be in parenthesis, as the authors' names are part of the discourse.
* Line 428: "Does pairing GPT-3 with MEM-PROMPT improves" - Typo. It should say "improve".

### Issues with the references

* When possible, please add URLs to the references as the template uses them.
* Johnson et al. (2017) – Cites preprint instead of the article's peer-reviewed version.
* Liu et al. (2021a) and Liu et al. (2021b) are duplicates of each other.
* Liu et al. (2021c) is missing the ArXiv ID.
* Marcus (2021) is missing the URL and the title is not clickable. The web page's URL should appear.
* Mitchell et al. (2021) is missing the ArXiv ID.

---

### Official Review · Reviewer_hT44 · 2022-03-24
**Simple and elegant approach to post-hoc error correction**

**Rating:** 7
**Confidence:** 4

**Review:**

### Paper Summary:

This paper focuses on improving GPT-3's performance post-deployment, without any retraining, via a growing repository of interactive user feedback. Through correcting GPT-3's misunderstanding of question intent via a key-value store of user questions and corrective feedback, the authors develop a system to edit prompts through such feedback from previously-asked, similar questions.

Evaluating on 4 tasks (lexical relations, word scrambling, and 2 variations of ethics reasoning), the authors show that their method of maintaining a growing memory store coupled with dynamically injecting feedback into prompts is useful in improving GPT-3's accuracy over time.


### Paper Strengths:

This paper takes a simple but effective step towards post-deployment error correction. Given that retraining (or, sometimes even large scale finetuning) may not always be tractable, the authors' conceptual framework of a lookup table for previously committed errors is straightforward and task-independent.

In addition, incorporating direct user feedback in future model interactions helps to improve interpretability of model output and the model's usability, given that small errors in intent understanding can be corrected post-hoc.


### Paper Weaknesses:

1. Evaluation of feedback: Given that the prompt is directly edited using feedback provided by users, it would be helpful to understand the model's sensitivity to the user feedback. For example, analysis of lexical sensitivity, robustness to noise in feedback, or other such analysis of user-provided feedback that did not aid accuracy/performance would help to understand the practical implications of using this framework with GPT-3.
2. Evaluation of "u": Likewise, for more complex questions or tasks, it seems like a more thorough evaluation of the generated question intent (via something like a human evaluation study) would be useful. Given that this approach is using humans in the loop, robust evaluation of the model's “understanding” of the task and the sensitivity/role of user feedback would help contextualize the limitations or practical applications of the approach.
3. Scalability: The key-value store (and thus the retrieval component) plays an instrumental role in the performance of the overall system design, but, to my knowledge, the paper does not include a discussion of the scalability of their approach. Given that the memory is simply expected to accumulate over time, this feels like an important dimension of analysis or discussion.


### Overall assessment

Overall, I think this paper is a nice step towards post-hoc correction of models with humans in the loop, and could be incredibly effective in certain practical settings.

### Typos

1. Line 176: add "than the"
2. Line 428: "improves" --> "improve"

---

### Decision · Program_Chairs · 2022-03-28

Accept